# Network Analysis of Seed Flow, a Traditional Method for Conserving Tartary Buckwheat (*Fagopyrum tataricum*) Landraces in Liangshan, Southwest China

**Yingjie Song [1], Qiong Fang [1], Devra Jarvis [2,3], Keyu Bai [4], Dongmei Liu [5], Jinchao Feng [1,*] and Chunlin Long [1,6,7,*]** 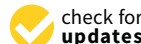

1   College of Life and Environmental Sciences, Minzu University of China, Beijing 100081, China
2   Bioversity International, Maccarese, 00057 Rome, Italy
3   Department of Crop and Soil Science, Washington State University, Pullman, WA 99164, USA
4   China-Bioversity International, Room 611 Old Building, Institute of Agricultural Resources and Regional Planning, Chinese Academy of Agricultural Sciences, Beijing 100081, China
5   State Environmental Protection Key Laboratory of Regional Eco-Process and Function Assessment, Chinese Research Academy of Environmental Sciences, Beijing 100012, China
6   Key Laboratory of Ethnomedicine (Minzu University of China), Ministry of Education, Beijing 100081, China
7   Kunming Institute of Botany, Chinese Academy of Sciences, Kunming 650201, China
*   Correspondence: fengjinchao@muc.edu.cn (J.F.); long.chunlin@muc.edu.cn (C.L.)

**Abstract:** The Yi ethnic group in Liangshan Prefecture, Sichuan Province in Southwest China have cultivated Tartary buckwheat (*Fagopyrum tataricum*) for at least a thousand years. Tartary buckwheat landraces are maintained through their traditional seed system. Field work and social network methodologies were used to analyze the seed sources and their flows, and to create a seed network map. Self-saving, exchanging with neighbors and relatives, and purchasing from the market were the main means farmers used to save and exchange Tartary buckwheat seeds. The flow of seed within villages was higher than between villages. Wedding dowry was an important pathway for seed flow among all of the villages. Of the 13 Tartary buckwheat landraces maintained, four landraces were exchanged frequently. The seed exchange network structure was affected by the number of Tartary buckwheat landraces, the age of nodal households, geographic environment, culture, and cultural groups. Nodal households play an important role in the conservation and on-farm management of Tartary buckwheat landraces.

**Keywords:** seed system; social seed network; on-farm conservation; farmers; Tartary buckwheat landraces

## 1. Introduction

Farmer seed exchange networks allow for the transfer of domesticated or undomesticated plant seeds via farmer-to-farmer gifting, swapping, bartering, or purchase, as well as trade or sale, which occurs outside of the commercial seed sector and formal seed regulation [1]. These methods are classified as informal seed systems, as contrasted from formal seed production within companies or public institutions [2]. Over 80% of smallholder farmers in developing countries depend on informal seed systems for their seed supply [3]. Farmer seed exchange networks are important not only for rural people's livelihood, but also for on-farm conservation of crop diversity and landraces' evolution [4].

Farmer seed exchange networks are an important method for the on-farm conservation of crop landraces, allowing for the continued adaption and evolution of crops to changing conditions [5–8].

The circulation of seed among farmers is central to agrobiodiversity conservation and dynamics. Crop varieties are often the result of the work of selection and exchange by generations of farmers and farming communities [9]. Seed exchange is commonly built upon trust, with such seed systems embedded in a pre-existing social structure and dependent upon farmers' social identity [10,11]. Social seed network analysis can help to understanding the factors that contribute to, or limit, the maintenance of diversity [12]. A network is formally defined by a set of nodes and a set of edges connecting those nodes together. Farmers represent the nodes in the network, and their seed exchanges represent the edges. Network centrality analysis can help us in understanding the dynamics and structure of the social seed network [13].

Local farmers use different channels to exchange seeds. Seed exchanges typically occur between relatives, neighbors, or friends, usually within the same ethnic group [14]. Often seed exchange and circulation takes the form of gifts. In small-scale farming communities, the circulation of crop landraces is often determined by marriage networks [15,16]. Marriage prohibitions and the exchange of seed through marriage influence the movement of seeds among villages, thereby shaping crop diversity at local and regional levels [17]. Kinship is another channel to exchange seeds. It determines the connectivity of farmer populations by favoring or limiting exchanges between communities [17]. In addition, kinship determines the connectivity of crop and livestock populations [18,19]. Marriage and kinship systems are often dependent on indigenous knowledge and culture, which influence the flow of seed among farmers, creating culturally defined agricultural environments that are akin to other environments occupied and used by humans [18].

Farmers also may choose to cultivate new varieties or seek alternative seed supplies. Such decisions often occur when farmers encounter disasters, which may be of a personal (e.g., poor health and individual production failure) or more general (e.g., floods, drought, and war) nature, creating an increased demand for off-farm seeds [19]. In these circumstances, farmers may acquire seeds via cash transactions, barter, as gifts, by exchanging one variety of seed for another, as a loan to be repaid upon harvest, or even by theft from another farmer's field [20–22]. Farmer seed networks can provide quality planting materials [23], increasing the resilience and autonomy of small-scale farmers, while reducing dependence on commercial seeds [24,25]. The role of farmer seed systems is central to the current debates on seed sovereignty and the conservation of crop germplasm [26].

Southwest China is famous for its rich biological and cultural diversity. It has been regarded as a global biodiversity hotspot [1]. According to our recent field surveys, most traditional seed systems have disappeared. Only in a few remote ethnic communities in southwest China do a very limited number of seed systems continue to be adopted by the local people [27,28].

Tartary buckwheat (*Fagopyrum tataricum*) is a minor crop based on global distribution, but it serves as a one of the staple foods for the Yi people in Liangshan [29]. The Yi people value Tartary buckwheat as a dietary staple for livestock feed, as well as part of their creation of myths and festivals [30]. A Yi proverb states that buckwheat is the mother of all crops [31,32]. Complicated climate and topographical differences in Southwest China increase the frequency of genetic exchanges between local wild and sibling species, cultivars and wild relatives, and within cultivars. Cultural customs, religious traditions, dietary habits, and other customs of the Yi affect local farmers' agricultural activities, which provide a cultural motivation for the conservation of agrobiodiversity. Above all, natural and human factors enrich the local seed systems, and guarantee that multiple varieties of crop landraces are cultivated. With the development of the modern economy and the erosion of traditional culture, farmers' seed exchange networks are being lost. If these networks cannot be conserved, they will disappear rapidly.

The purpose of this study was to analyze the dynamics that structure social networks that influence seed exchange, as well as how these dynamics impacts on Tartary buckwheat's on-farm conservation. In order to understand the practices of on-farm conservation and seed circulation, we investigated traditional practices of on-farm conservation in Liangshan of Sichuan Province in Southwest China. We documented the flows of Tartary buckwheat seed within and between communities, we characterized the seed networks in three villages, we analyzed the landraces in the

seed exchange network, and we analyzed the factors influencing farmers' seed exchange. We suggest that these findings are applied so as to facilitate on-farm conservation and breeding development.

## 2. Materials and Methods

### 2.1. Study Sites

The Liangshan Yi Autonomous Prefecture is located in the upper Yangtze River watershed of southwest Sichuan Province, near the Hengduan Mountains. It is the largest community of Yi people in China, with a population of 2.64 million. The community has a long and rich folk culture. For several reasons, Liangshan is a poverty-stricken area, and has been designated as an important region for poverty reduction by Chinese authorities. Study sites were selected from Zhaojue and Meigu counties of Liangshan (Figure 1). Four districts were identified based on their importance in Tartary buckwheat production and consumption; all of the districts were located at an altitude of greater than 2050 m (Table 1 and Figure 1). Four to six villages were selected based on their resources for cultivating Tartary buckwheat, giving a total of 21 villages across all four districts. In addition, three villages were selected from District A for an additional case study (Figure 1). Some background of the study sites is provided in Table 1.

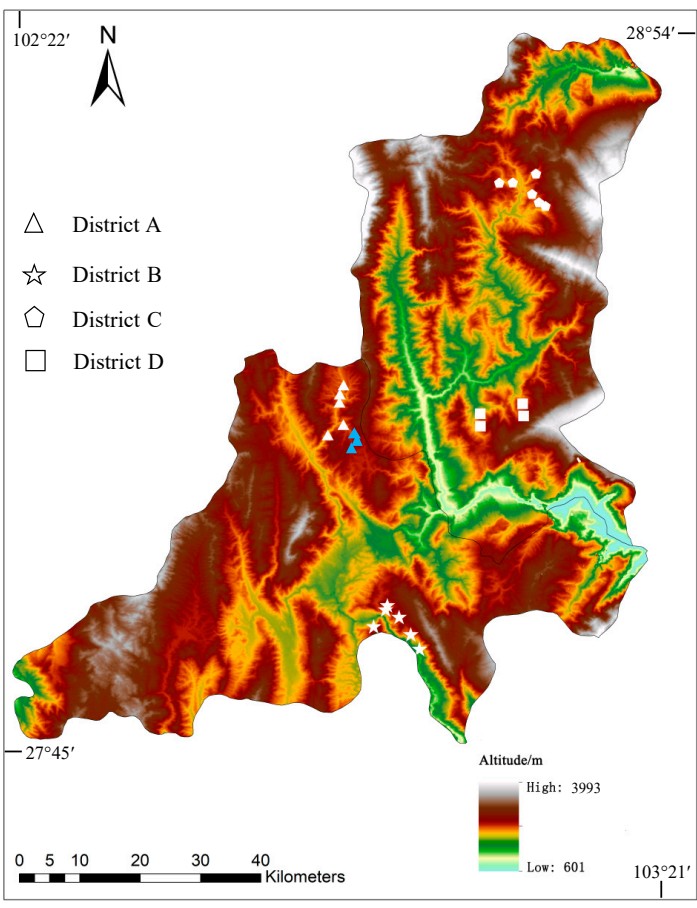

**Figure 1.** Study sites. Fieldwork was carried out in 21 villages in four districts within Zhaojue and Meigu counties. Different shapes of sites in the map represent the four districts. Blue triangles are another three villages that were selected for an additional case study.

**Table 1.** Characterization of study sites for analyzing social seed network, Liangshan, China.

| District | NO. | Sample Site | Longitude | Latitude | Altitude (m) | Households Interviewed |
|---|---|---|---|---|---|---|
| A | 1 | Gemowaxi | 102.84 | 28.22 | 2280 | 9 |
| | 2 | Kumo | 102.86 | 28.27 | 2370 | 9 |
| | 3 | Hajue | 102.87 | 28.28 | 2380 | 10 |
| | 4 | Juega | 102.86 | 28.30 | 2400 | 8 |
| | 5 | Wazhajiagu | 102.86 | 28.24 | 2390 | 10 |
| B | 6 | Hexi | 102.91 | 27.94 | 2360 | 8 |
| | 7 | Tangqie | 102.97 | 27.90 | 2050 | 9 |
| | 8 | Tubizili | 102.96 | 27.93 | 2460 | 11 |
| | 9 | Luogu | 102.94 | 27.95 | 2530 | 8 |
| | 10 | Aweiluoha | 102.93 | 27.97 | 2520 | 9 |
| | 11 | Taodu | 102.92 | 27.96 | 2360 | 9 |
| C | 12 | Ejue | 103.15 | 28.57 | 2540 | 10 |
| | 13 | Ejueerzu | 103.16 | 28.57 | 2540 | 8 |
| | 14 | Ema | 103.14 | 28.58 | 2430 | 10 |
| | 15 | Yideamo | 103.15 | 28.61 | 2440 | 9 |
| | 16 | Gandu | 103.11 | 28.60 | 2270 | 7 |
| | 17 | Cainaijian | 103.09 | 28.60 | 2250 | 9 |
| D | 18 | Yise | 103.13 | 28.27 | 2470 | 7 |
| | 19 | Wanigu | 103.13 | 28.25 | 2520 | 9 |
| | 20 | Yideamo | 103.07 | 28.27 | 2470 | 8 |
| | 21 | Yiluoerhe | 103.07 | 28.25 | 2080 | 10 |
| Case study | 22 | Juetuo | 102.86 | 28.20 | 2400 | 22 |
| | 23 | Anqule | 102.86 | 28.20 | 2200 | 13 |
| | 24 | Wazachongle | 102.86 | 28.21 | 2390 | 20 |

## 2.2. Household Seed System Survey

We conducted fieldwork in 2017 and 2018 to investigate Tartary buckwheat landraces and seed exchange networks. Approximately ten households per village were surveyed (Table 1 and Figure 1) via participatory rural appraisal (PRA) using key informant interviews and semi-structured interviews [33–36]. As mentioned above, three of the villages (22, 23, and 24; Table 1 and Figure 1) from District A were selected for a case study, for the purpose of analyzing network centrality data. During this case study, we lived and worked with the local farmers to learn their customs, indigenous knowledge, and seed exchange networks. In total, including the case study, we surveyed 242 households (Figure 2) across the 24 villages.

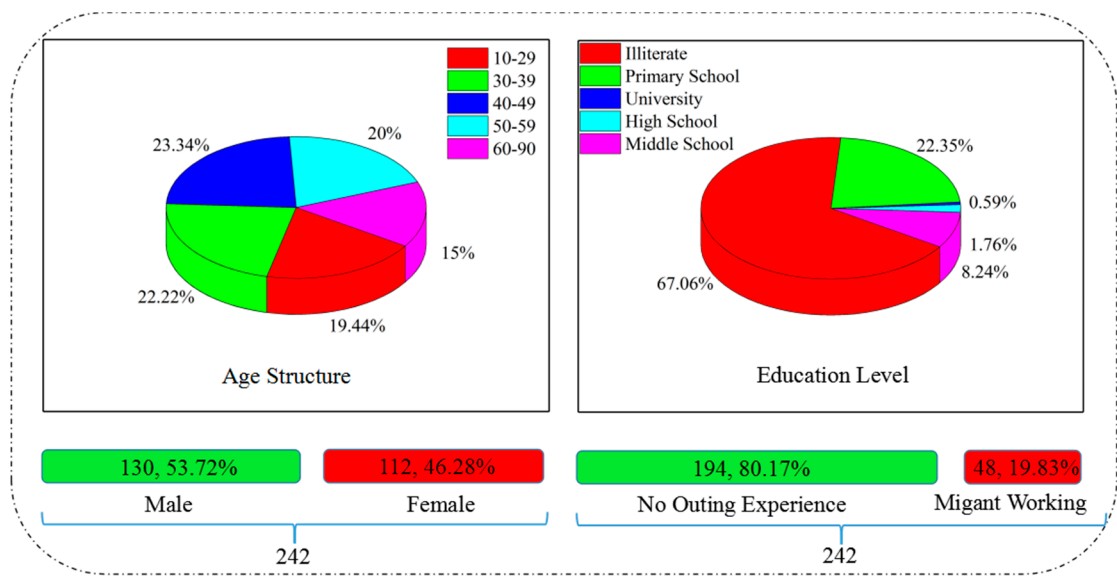

**Figure 2.** The demographics.

All of the 242 households were given a questionnaire that covered several topics, including Tartary buckwheat landraces, seed management practices, seed sources, and seed flow (Supplementary S1). In addition, the fifty-five households from the case study (villages 22, 23, and 24) were interviewed for seed source and flow data, which was used for the network map development.

### 2.3. Data Analysis

The seed source data were analyzed to characterize the seed flows within and between villages, as well as between different people. The exchange count was defined as the total number of seed exchanges, summed across all 242 households. The seed network data analysis and mapping were performed using UciNet, version 6.21 [37]. These analyses resulted in three main measures of network centrality, namely: degree centrality, closeness, and betweenness centrality [38].

The degree centrality of a household measures its level of direct connectedness to other households; a higher number indicates greater connectivity. The closeness centrality of a household measures its minimal distance in the network to all other households; a smaller number indicates that it has more direct connections to other households. The betweenness centrality of a household is a measure of the number of shortest paths that go through the household; a high value indicates more indirect connections via these shortest paths.

Furthermore, we define a nodal household as a household with a degree greater than five and connected to at least three other households. A connector household has degree less than five and is connected to at least two other households. Finally, Pearson's R correlations were computed pairwise between the number of landraces per household, the age of household, and the centrality degree using SPSS 20.0 software (SPSS, Chicago, IL, USA).

## 3. Results

### 3.1. Seed Sources

Self-saving, exchanging with neighbors and relatives, and purchasing from the market were the main methods used to acquire Tartary buckwheat seed across all of the villages. Most of the households interviewed used their own seeds or exchanged with neighbors and relatives (Figure 3). One hundred and seventy-seven households exchanged seeds with neighbors, while only 25 households obtained seeds from the market. In Districts A and C, almost all of the households used seeds obtained from relatives and neighbors, and only a small number of households obtained seeds from the market. In Districts B and D, only two households each obtained seeds from the market. Figure 2 illustrates that the sources of Tartary buckwheat seeds used by households are various. A higher proportion of seeds were obtained within villages, rather than from outside the village. Within the villages, the majority of households indicated that they obtained seeds through exchanging with non-relatives. One third of the households obtained seeds through exchanging with their immediate relatives. About 33% of households obtained seeds through borrowing. Only 15 households purchased seeds from markets. Between villages, it was observed that the seed sources were primarily exchanged with relatives, which relates to marriage. Fifty-three households accessed seeds through exchange with non-relatives, and 35 households through borrowing from friends. Another important pathway for obtaining Tartary buckwheat seed is through marriage dowry. Only 23 households cultivated agriculture bureau seeds.

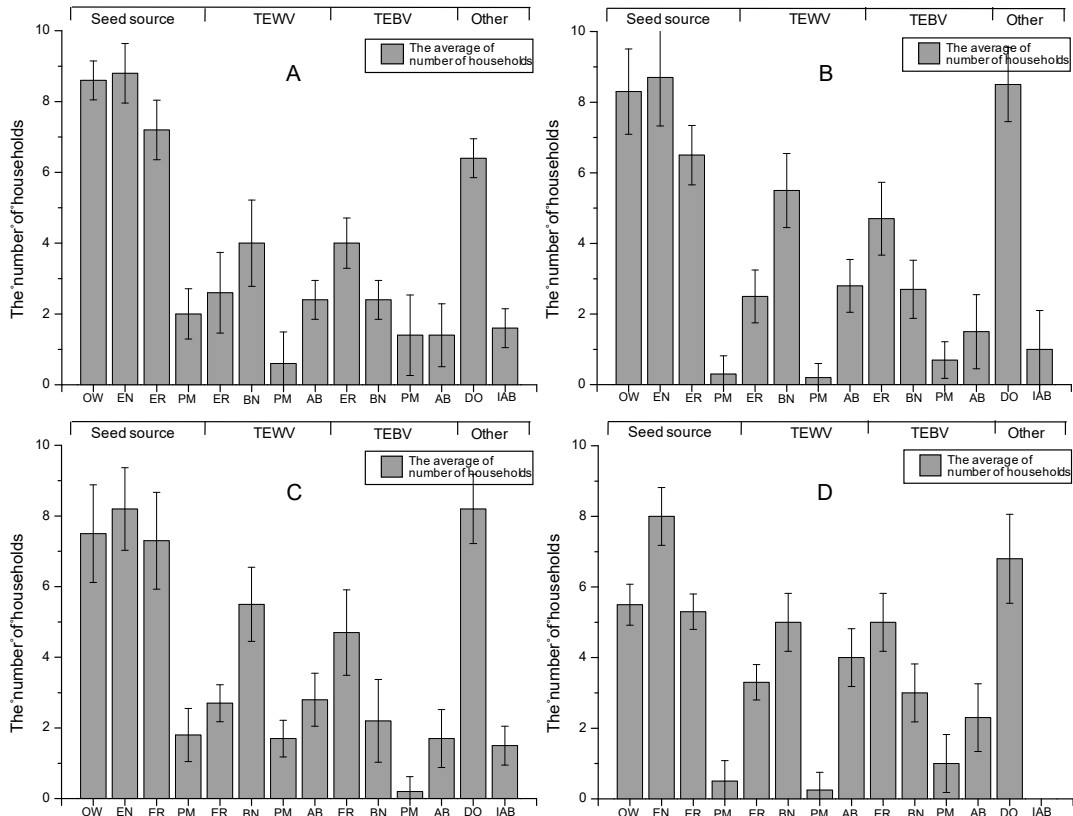

**Figure 3.** Tartary buckwheat seed sources. The figure shows seed source, type of exchange within village (TEWV), and type of exchange outside village (TEBV). OW—saving-self; EN—exchanging with neighbors; ER—exchanging with relatives; PM—purchasing from market; DO—marriage dowry; IAB—issued by agricultural bureau. A, B, C and D represent four different districts.

### 3.2. Flows and Sources of Tartary Buckwheat Landraces

In the 21 villages surveyed in the study area, 13 Tartary buckwheat landraces were recorded. The household demand for Tartary buckwheat landraces is illustrated in Table 2. Farmers indicated that they obtained seeds of landraces numbered 1, 2, 3, and 4 from their relatives, neighbors, and from marriage dowries. From the exchange count of these four landraces, we conclude that these landraces are the most popular ones. The second popular are landraces 5, 6, 7, and 8; the frequency of seed exchange is much lower than the most popular landraces. According to the farmers, landrace 6 is exotic or hybrid, so it was obtained through the agriculture bureau. The exchange count of landraces 9, 10, 11, 12, and 13 were the lowest. Farmers used multiple seed sources and cultivated multiple varieties. The demand was dominated by the most popular four landraces (1, 2, 3, and 4), and 66% of the interviewees exchanged these landraces with neighbors and relatives within the village. About 35% of interviewees exchanged seeds with relatives outside of their own village. Approximately 5% of interviewees obtained the most popular varieties from markets. About 30% of interviewees exchanged seeds through marriage dowry. No interviewee obtained the most popular landraces from the agriculture bureau. The flow of Tartary buckwheat landraces was primarily between neighbors, relatives within villages, and relatives outside villages (Table 2).

**Table 2.** Tartary buckwheat landraces seed source.

| | | | \multicolumn{6}{c}{Tartary Buckwheat Landraces as Seed Source by Household Number} | | | | | |
| No. | Landraces | Local Name | Within Village | | Outside Village | | Dowry | Agriculture Bureau |
| | | | Neighbors | Relatives | Relatives | Market | | |
|---|---|---|---|---|---|---|---|---|
| 1 | en g tumuer | $ŋɡɯ^{33}$ $tɕu^{33}$ | 86 | 67 | 93 | 11 | 56 | 0 |
| 2 | en g wu | $ŋɡɯ^{33}$ $vu^{33}$ | 77 | 53 | 58 | 6 | 41 | 0 |
| 3 | shuang lang g | $ʐuo^{33}$ $ɬɯ^{21}$ | 79 | 51 | 62 | 7 | 26 | 0 |
| 4 | ri g | $tshu^{33}$ $ŋɡɯ^{33}$ | 39 | 21 | 46 | 0 | 17 | 0 |
| 5 | en g nao | $ŋɡɯ^{33}$ $nuo^{33}$ | 11 | 6 | 15 | 5 | 0 | 3 |
| 6 | hai ga g | $hie^{33}$ $ɬɯ^{21}$ $ŋɡɯ^{33}$ | 19 | 6 | 15 | 3 | 0 | 20 |
| 7 | wo g | $wo^{33}$ $ŋɡɯ^{33}$ | 9 | 5 | 11 | 0 | 0 | 0 |
| 8 | en g jie | $ŋɡɯ^{33}$ $tɕɛ^{33}$ | 12 | 3 | 15 | 0 | 3 | 0 |
| 9 | en g chi | $ŋɡɯ^{33}$ $tɕhɿ^{21}$ | 5 | 1 | 4 | 0 | 1 | 0 |
| 10 | en g zhi | $ŋɡɯ^{33}$ $IA^{33}$ | 0 | 0 | 2 | 0 | 0 | 0 |
| 11 | en g la | $ŋɡɯ^{33}$ $pu^{33}$ | 0 | 0 | 3 | 0 | 0 | 2 |
| 12 | en g bu | $ŋɡɯ^{33}$ $tshu^{33}$ | 1 | 0 | 1 | 0 | 0 | 1 |
| 13 | en g chu | $ŋɡɯ^{33}$ $tʂhi^{33}$ | 0 | 0 | 2 | 0 | 0 | 0 |

Note: Local names are denoted by international phonetic symbols.

### 3.3. Analysis of Centrality Data

The seed network analyses were conducted based on household surveys from 55 households in three adjacent villages. Table 3 shows the characteristics of the households that were active in the seed network, and their nodal data including degree, closeness, and betweenness. It also shows the households' position in the network, such as a nodal farmer and connector farmer.

Households were characterized as nodal (NC) and connectors (C) in this study. There were 21 nodal households (NC; degree >5.00), defined as households linked to three or more other households. In addition, there were 19 connector households (C; degree <5.00), defined as households linked to two other households. Nodal households connected with other members of the network and occupied positions between networks. By linking together more households, connector and nodal households form a network for seed flow. Seven, five, and nine NC households were found in the three case study villages, respectively (GJ0270, JJ2472, and MS3640). Most interestingly, the age of the nodal households was older than that of the connector households. Other households were designated as access households, defined as households located in the end of one chain of network. For example, DT0135 and WG0748 in village #1 only obtained seed from other households, and did not exchange with others. Above all, the nodal and connector households were evenly distributed in each village, and made up a network for Tartary buckwheat seed flow.

**Table 3.** Characterization and nodal data of the household in seed network.

| Village | Household | Nodal Data Calculated on the Basis of Network Analysis | | | Position in Network |
| | | Degree | Closeness | Betweenness | |
|---|---|---|---|---|---|
| | DT0135 | 1.69 | 9.75 | 0.00 | - |
| | GJ0270 | 8.47 | 10.68 | 14.82 | NC |
| | QH0359 | 3.39 | 9.98 | 0.00 | C |
| | QB0442 | 1.69 | 9.75 | 0.00 | - |
| | WE0674 | 6.78 | 11.23 | 19.38 | NC |
| | WG0748 | 1.69 | 10.20 | 0.00 | - |
| | LG0837 | 1.69 | 10.20 | 0.00 | - |
| | FD0955 | 6.78 | 10.29 | 7.85 | NC |
| | GT1033 | 3.39 | 9.76 | 1.06 | C |
| | DZ1152 | 3.39 | 10.12 | 4.51 | C |
| 22 | QR1250 | 3.39 | 10.48 | 3.40 | C |
| | QR1370 | 5.08 | 11.02 | 5.04 | NC |
| | GB1458 | 3.39 | 9.70 | 1.12 | - |
| | AN1580 | 8.47 | 10.33 | 15.15 | NC |
| | QZ1653 | 3.39 | 10.74 | 9.99 | C |
| | MQ1737 | 3.39 | 9.51 | 6.07 | C |
| | AB1839 | 3.39 | 8.79 | 3.09 | C |
| | WN1970 | 3.39 | 10.86 | 10.19 | C |
| | LD2042 | 6.78 | 11.32 | 15.60 | NC |
| | GQ2168 | 3.39 | 10.70 | 9.68 | C |
| | MG2258 | 1.69 | 8.14 | 0.00 | - |
| | RX0548 | 6.78 | 11.80 | 25.18 | NC |
| | MG2363 | 5.08 | 11.45 | 12.61 | NC |
| | JJ2472 | 6.78 | 11.89 | 20.27 | NC |
| | WN2580 | 6.78 | 11.82 | 20.51 | NC |
| | WY2643 | 3.39 | 11.28 | 6.62 | C |
| | AH2762 | 3.39 | 10.53 | 1.03 | C |
| 23 | RL2858 | 5.08 | 10.84 | 7.09 | NC |
| | AD2924 | 3.39 | 11.02 | 2.30 | C |
| | QP3035 | 1.69 | 9.94 | 0.00 | - |
| | KB3135 | 1.69 | 9.94 | 0.00 | - |
| | EZ3237 | 3.39 | 10.63 | 6.18 | C |
| | MY3354 | 1.69 | 9.94 | 0.00 | - |
| | AJ3460 | 11.86 | 10.92 | 20.82 | NC |
| | LQ3575 | 1.69 | 9.94 | 0.00 | - |
| | MS3640 | 1.69 | 9.96 | 0.00 | - |
| | WL3764 | 6.78 | 10.94 | 8.87 | NC |
| | QR3850 | 3.39 | 10.26 | 0.00 | - |
| | EJ3966 | 1.69 | 9.72 | 0.00 | - |
| | WS4062 | 6.780 | 10.65 | 6.06 | NC |
| | BM4138 | 3.390 | 9.86 | 0.17 | C |
| | RL4247 | 6.780 | 10.94 | 4.90 | NC |
| | WB4340 | 3.390 | 10.00 | 0.27 | C |
| | RN4445 | 6.780 | 10.78 | 7.07 | NC |
| | SP4544 | 3.390 | 10.80 | 1.69 | C |
| 24 | GG4662 | 10.16 | 11.61 | 20.95 | NC |
| | AG4755 | 5.08 | 11.06 | 9.46 | NC |
| | WA4838 | 3.39 | 10.80 | 1.69 | C |
| | WQ4960 | 5.08 | 10.36 | 2.81 | C |
| | DR5066 | 3.39 | 10.72 | 8.57 | C |
| | ES5158 | 3.39 | 10.27 | 2.25 | NC |
| | ES5250 | 8.47 | 10.78 | 16.45 | NC |
| | KR5336 | 3.39 | 9.86 | 0.00 | - |
| | SR5470 | 5.08 | 9.88 | 3.08 | NC |
| | QY5543 | 1.69 | 9.07 | 0.00 | - |

Note: DT0135—the alphabets are abbreviation of households' name. The first two numbers are the serial number and the last two are the age of the households. NC—nodal households; C—connector households.

### 3.4. Analysis of Network Map

Based on the analysis of the network and centrality data, the network map was generated through UciNet. The information and location of the households provides a visual representation of the association between the households in the network and Tartary buckwheat landraces conservation (Figure 3). The color of the node indicates the number of Tartary buckwheat landraces owned by households. The size of the node shows the degree of the households. The number next to the node corresponds to the identification number of the households. The first two alphabetical characters are the name abbreviation. The last two numbers are the age of the household. The network was divided into three parts, according to the village boundary. As the previous results showed, seeds were exchanged mainly among neighbors and relatives within the village. The network had a low density between villages, indicating that there were few ties between relatives who exchanged seed (Figure 4).

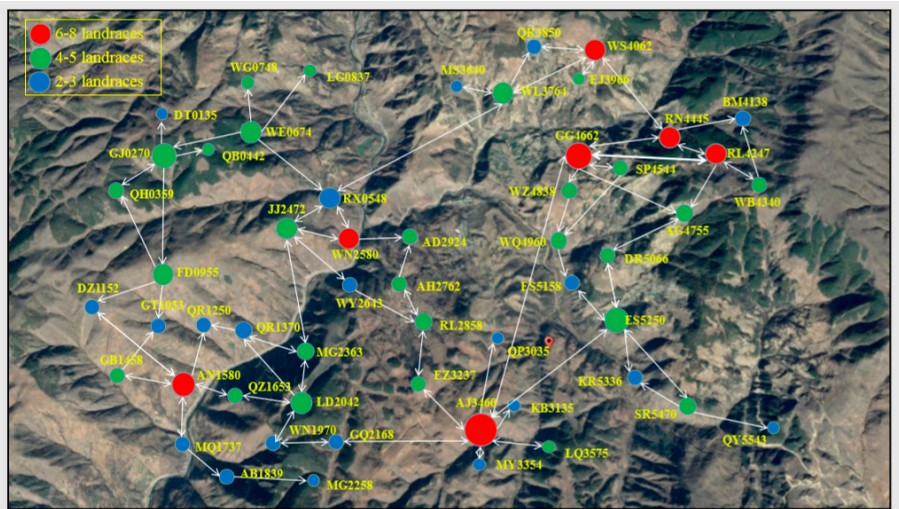

**Figure 4.** Network map. It shows the Tartary buckwheat seed flow between three villages in Zhaojue County (55 households). The color of the nodes indicates the number of Tartary buckwheat landraces retained by households, the size of the nodes is an indication for the degree of the households, arrows indicate the direction of the seed flow, and the location of the nodes is the actual location of households. Lastly, the network is not closed.

## 4. Discussion

### 4.1. Structure of Seed Exchange Network and Flows of Landraces

Tartary buckwheat seeds are primarily sourced and circulated via farm-saving and exchanging with neighbors and relatives. The seeds flow more frequently within villages than between villages. Seed flow between villages is primarily carried out through relatives. Another important pathway for seed flow is marriage dowry. Some studies have suggested that seed exchange is not the main mechanism for seed acquisition, as households only exchange seeds occasionally [39–41]. In contrast, our research showed that households obtained seed mainly through farm-saving and seed exchange. Purchasing seed from markets accounted for a low proportion of seed acquisition. Seed flow between farmers affects the diversity of Tartary buckwheat. Most studies on farmer seed exchange systems indicate that a high level of exchange between neighbors or family members has a positive impact on the flow of the genetic diversity of crops [14,42]. Vigouroux and Delêtr reported that seeds are often inherited via gifts or dowry at weddings in developing countries [18,43]. Also, in matrilineal societies, the continuous inflow of manioc landraces through marriages networks contributes to increasing varietal diversity at the community level [17]. In small-scale farming communities, marriage networks are often used as a channel for the circulation of crop landraces [16]. In our study, almost

all of the households obtained or supplied Tartary buckwheat landraces through marriage dowry. In addition, even after the wedding, households still continued to exchange seeds. Households joined by marriage, therefore, influence the movement of Tartary buckwheat landraces, helping to maintain Tartary buckwheat diversity at a local and regional level, as well as to conserve crop genetic diversity on-farm.

The low adoption of Tartary buckwheat varieties issued by the agriculture bureau can be explained by these varieties having poor environmental adaptability. Despite the lower yields available, households often choose to continue cultivating or exchanging local landraces, because the yields are stable and predictable, and their consumption characteristics are well known [18]. The high yield and environmental adaptability may be a result of seeds that have adapted well to local conditions after generations of farmers' selection, with regular seed exchanges leading to yield stability [44,45].

Our study shows that local farmers continue to cultivate Tartary buckwheat landraces, obtaining seeds through traditional exchange practices. There are 13 Tartary buckwheat landraces flowing among the households, with four landraces being the most popular. The local farmers expressed satisfaction with their seeds and crop yield. Taking landrace 1 as an example, farmers evaluate it as the most trustworthy landrace because its hardiness, drought resistance, good yield, high altitude plantation, and high nutritional value. It can be concluded that seed exchange is an important usage of traditional knowledge, and that it enriches the diversity of Tartary buckwheat landraces.

*4.2. Centrality Analysis and Tartary Buckwheat Landraces Conservation*

Through a centrality analysis of the seed network, we can understand the patterns of exchange and flows of seed within and between villages. Our results suggest that, at a household level, measures of network centrality are associated with families that conserve Tartary buckwheat landraces. These central or nodal families retained four to eight Tartary buckwheat landraces, and many also have a rich traditional knowledge. Other studies have found that centrality in the seed exchange network is associated with local landraces conservation and knowledge, reinforcing previous findings on the importance of seed exchanges in ensuring the maintenance of local agrobiodiversity [44,46]. Households in Liangshan are connected in complex seed exchange networks [47,48]. Understanding the households' position in seed networks can be useful for the design of intervention strategies targeting conservation [38].

Nodal households in the seed exchange network are vital for maintaining crop diversity on-farm, and for managing related knowledge [49]. An example is household AJ3460, an NC household that links seven others households and two other villages. This household retained eight Tartary buckwheat landraces. Our survey also showed that this household had significant knowledge of the characteristics of each of the eight landraces. The household obtained these eight landrace seeds from outside the village, and then supplied the seeds to the other households within their own village. In addition, they also obtained the seeds from within the village, and then supplied it to households in other villages. These nodal households are essential for maintaining diversity of seed exchange both within or outside of villages. Supporting or training these nodal households on seed cleaning and multiplication could enhance crop genetic diversity [38]. Seed exchange network analysis could be an efficient method for on-farm genetic diversity management by supporting Tartary buckwheat landraces to flow within the community.

*4.3. Factors Affecting the Structure of Seed Exchange Networks*

From the network map, we could identify the level of some dense links between households, and the main factors impacting the seed exchange network. First, the number of landraces contained by households and the centrality degree are positively correlated (Pearson's r = 0.73; $p < 0.01$). This indicates that the more landraces retained by households, the more important the household's role will be in the network. Second, although the age of the nodal household is not correlated with the centrality degree (r = 0.06), all of the nodal households' age is over 42. This suggests that age plays an

important role in the network. As Figure 2 shows, village 2 is located in the valley, whereas villages #2 and #3 are located on the hillside and hilltop. Village #2 connects villages #1 and #3. However, between villages #1 and #3, there is not seed exchange. Thus, the terrain of the village also affects the network structure. Elders from villages #1 and #3 told us that it was convenient to exchange seeds with village #2, because they did not want to climb another mountain for exchanging seeds. Furthermore, they got a high yield from the Tartary buckwheat seeds that came from village #2. This is similar to a study by Chambers et al. [45], which showed how both the physical and human geography of landscapes influence the varieties of maize grown, and thus affect the seed acquisition practices. The impact of culture and ethnic groups on seed exchange networks has been widely documented. Confirming this, our study found that culture and cultural groups influenced the seed exchange and diversity of Tartary buckwheat landraces. For example, households RX0548, AN1580, and GG4662 are the *bimo*, a shaman responsible for hosting various rituals. As professional ritualists of the Yi ethnicity, the *bimo* is a high-position religious specialist, who is a master of the ancient Yi language scriptures, including astronomy, calendar systems, epics, and medicine in their village. Their centrality degrees were 6.87, 8.47 and 10.16, indicating that their position in the network is important. They master the culture of the Yi people, and they know about the uses of Tartary buckwheat in rituals. They even plant a small amount of multiple Tartary buckwheat landraces, supplying it to the other households, along with management knowledge.

## 5. Conclusions

In this study, we have revealed four main types of seed sources for Tartary buckwheat, namely: self-saving, exchanging with neighbors, exchanging with relatives, and purchasing from the market. Four of the 13 landraces were the most popular, flowing among the households frequently. Seed network analysis was used to identify the households who play different roles in the seed exchange system. An analysis of the centrality and the network map showed that there were 21 households that played nodal roles in the network. These household both retained four to eight Tartary buckwheat landraces, and had rich traditional knowledge of the characteristics and management of buckwheat landraces. To support the conservation of Tartary buckwheat landraces, training on seed cleaning and seed management should be provided for the nodal households. Combining fieldwork with network analysis, we concluded that the seed exchange network structure is impacted the most by the number of Tartary buckwheat landraces, the age of nodal households, geography, and culture and ethnic groups. Seed exchange networks help farmers meet their production and cultural needs. Understanding these networks can provide guidance to local governments to support the maintenance of local traditional seed exchange systems, teaching locals how to select seeds and reducing interference from modern varieties. Understanding seed networks and social culture can contribute to increasing seed security and resilience to current and future environmental change.

**Supplementary Materials:** The following are available online at http://www.mdpi.com/2071-1050/11/16/4263/s1. Supplementary S1: Questionnaire information.

**Author Contributions:** C.L.L. and J.C.F. conceived this study. Y.J.S. carried out the field investigations, analyzed and interpreted the data, and drafted the manuscript. Q.F. and D.M.L. participated in the data analysis. D.J. and K.Y.B. edited the English and provided useful comments. C.L.L. finalized the manuscript. All of the authors read and approved the final manuscript.

**Funding:** This research was funded by the National Natural Science Foundation of China [31761143001, 31870316], the Ministry of Ecological Environment of China (2019HB2096001006), the Key Laboratory of Ethnomedicine (Minzu University of China) of the Ministry of Education of China [KLEM-ZZ201806, KLEM-ZZ201906], Minzu University of China [Collaborative Innovation Center for Ethnic Minority Development and yldxxk201819], and the Ministry of Education of China and State Administration of Foreign Experts Affairs of China [B08044].

**Acknowledgments:** Eric Miller from the University of California at San Francisco carefully edited the English. Financial support for this study came from National Natural Science Foundation of China (31761143001, 31870316), the Ministry of Ecology and Environment (2019HB2096001006), the Key Laboratory of Ethnomedicine (Minzu University of China) of the Ministry of Education of China (KLEM-ZZ201806, KLEM-ZZ201906), Minzu University of China (Collaborative Innovation Center for Ethnic Minority Development and yldxxk201819), and the Ministry

of Education of China and State Administration of Foreign Experts Affairs of China (B08044). We are very grateful to the Yi people who provided their valuable information, traditional knowledge, and culture about Tartary buckwheat.

**Conflicts of Interest:** The authors declare that they have no competing interest.

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
