# Peer review of "Network Analysis of Seed Flow, a Traditional Method for Conserving Tartary Buckwheat (Fagopyrum tataricum) Landraces in Liangshan, Southwest China"

_sustainability, doi:10.3390/su11164263_

Round 1
Reviewer 1 Report
General and specific comments:
1. The manuscript is of merit and interest as it provides significant and useful data about seed networks (seed sources and their flow) in several villages of Sichuan Southwestern China, particularly the case of buckwheat landraces, providing significant information to contextualize the local uses of such species and to understand how seed exchange networks and diversity is affected by different factors, e.g. local culture, villages, type of farmers... Moreover such information is fundamental to develop strategies in situ for landraces conservation, improving on-farm conservation, agrobiodiversity and landraces seed security.
2. The theme is scientifically sound; the title and abstract introduce readers to general context and main objectives of the paper. However, the study area and the methods used to perform the surveys are not satisfactorily described. It is not easy to understand how the authors surveyed the villages. Table 2 summarizes data from 21 villages, but in line 144 it is said«…Three villages were selected». That is to say that the section “household seed system survey” should be better described in order to provide a general picture of the field work. Moreover it seems that Table 2 also includes some results so these data should be in the Results section.
Figure 1 and Figure 2 need legends to be autonomous.
3. Results section provides important and extensive data on the main topic of the paper but some information is codified, shortly presented, sending to a Figure or Table, which does not make easy for readers to understand the main data.
Results section denotes that authors’ approaches were not sufficiently detailed and explained before in methods section. The structure of this section could be improved in order to emphasize the documented data and to make the topic much more attractive to readers. This section is a bit dense writing. Figure 2 is not easy to analyze. Moreover the districts A,B, C and D from Figure 1 could be also marked in Figure 2 providing a better idea of the changes outside each district.
4. Discussion and conclusions focus the major findings emphasizing importance of seed networks and local knowledge to improve seed security and biodiversity conservation strategies. These sections are much clearer than the methods and results sections.
5. The literature cited seems pertinent.
6. The manuscript needs a general revision of English style and language. Some ideas are not quite clear and it might be due to style and language. For instance: Lines 70, 99; 233; 238-239; other…
Author Response
We are very grateful to your kind comments. Here is the point-by-point response to your comments:
1. The study area and the methods used to perform the surveys are not satisfactorily described.
Information of three villages for case study were added in Table 1. It was distinguished from other 21 villages. Table 2 should be moved to the section “Results”, and it was replaced with a new figure.
Legends of Figure 1 and Figure 2 had been added.
2. Result section is a bit dense. Figure 2 is not easy to analyze. The district A, B, C and D from Figure 1 could be also marked in Figure 2 providing a better idea of the changes outside each district.
Result section had been revised, and some discussions were deleted.
In the method section, we pointed that in order to analyze centrality data and network system developing other three adjacent villages (22, 23, 24) (Table 1 and Figure 1) from district A were selected for case studying. These three village were also marked in Figure 1. In Figure 2, the network diagram showed the seed flow among these three villages.
3. Some ideas are not quite clear and might be due to style and language.
Lines 70, 99, 108, 117, 151, 233, 238, 362, 371, 406 and 413 were corrected.
Thank you again!
Reviewer 2 Report
In this interesting manuscript, Authors develop a seed flow network map for Tartary buckwheat among 4 districts in Liangshan region of China. Authors try to understand the factors underlying the maintenance and conservation of seed diversity, flow and exchange both within and among communities to inform the management and conservation of the buckwheat landraces. The results presented herein will be of considerable interest to the conservation community.
Specific Comments:
While the large portion of the manuscript is clearly written, in several sentences grammatical mistakes are present. These interfere with the readers and reviewers ability to correctly understand the import of what authors are trying to say. A professional proofreading and corrections of these language errors will greatly benefit the manuscript. Ex. sentences -- Line 108, 117, 151, 406,413, 371, 362
Line 213: UciNet software's publication should be cited
The survey questions used for the study should be provided as supplemental material. If possible, the survey response data should also be made available as suplemental material.
Table 2 is very hard to decipher at a glance, authors will do better to present this information in barplots or boxplots with mean and range for each District, which is a much better visual representation for this data.
Please provide proper legend for Table 3, Table 4 and Figure2
The household ID nos. in Figure2 shoudl be provided in the column 2 - "Household ID no." in Table 4. This will make the information provided much clearer.
The age of household is mentioned again and again in manuscript but the materials and methods omit to define how this age is calculated.
Line 57: Local farmers use "main channels" to exchange seed. What is meant by Main channels?
Table 3 : Please define exactly how the numbers for "Demand" in this table are arrived at? Preferable in a Table legend that is currently missing
Line 245: Please define the term "exchange frequency" in methods section
Kine 315: "The seed flow between farmers affects the diversity of Tartary Buckwheat" -- This conclusion has no statistical signifcance support in the present manuscript. Perhaps the authors can add such support.
Author Response
Thank you indeed for your helpful comments. We are now providing a point-by-point response to your comments:
1. A professional proofreading and corrections of these languages errors will greatly benefit the manuscript.
We are very sorry for the incorrect writing. Lines 70, 99, 108, 117, 151, 233, 238, 362, 371, 406 and 413 have been corrected or rewritten.
2. UciNet software’s publication should be cited.
We have added the publication of UciNet in reference section.
3. The survey questions used for the survey should be provided as supplemental material.
We have added Supplementary S1 about survey questions. But the response data is too large to be provided here.
4. Table 2 is very hard to decipher at a glance.
Thanks for the suggestion. We have made a histogram to replace Figure 2.
5. Please provide proper legend for Table 3, Table 4 and Figure 2.
We are very sorry for our negligence of legends for tables and figures. We have added legends to all tables and figures.
6. The household ID in Figure 2 should be provided in the column 2 in Table 4.
The households in Table 4 are from three case study villages (22, 23, 24), and different from the household from other 21 villages. The centrality data and network diagram data were from three case villages.
7. How age is calculated?
We have illustrated that the information of household was from fieldwork in the methods section. A correlation analysis between the number of landraces of household, the age of household and centrality degree were conducted using SPSS.
8. Line 57: Local farmer use “main channels” to exchange seed. What is mean by Main channels?
We are sorry for our incorrect using. What we mean are “Local farmers use different access to exchange seed”.
9. Please define exactly how the numbers for “Demand” in Table 3 are arrived at?
We are very sorry for adding redundant implication in this table. It was changed to be “Tartary buckwheat landraces seed source”.
10. Please define the term “exchange frequency” in methods section.
We have added a definition according to the reviewer’s comment, and defined exchange frequency as the number of households using different accesses to exchange seed.
11. “The seed flow between farmers affects the diversity of Tartary buckwheat” This conclusion has no statistical significance support in manuscript.
Thanks a lot. We did not have enough data to support this conclusion. Therefore we deleted it.